# The Macroalgal Biostimulant Improves the Functional Quality of Tomato Fruits Produced from Plants Grown under Salt Stress

**Kanagaraj Muthu-Pandian Chanthini** [1], **Sengottayan Senthil-Nathan** [1,*], **Ganesh-Subbaraja Pavithra** [1], **Arul-Selvaraj Asahel** [1], **Pauldurai Malarvizhi** [1], **Ponnusamy Murugan** [1], **Arulsoosairaj Deva-Andrews** [1], **Haridoss Sivanesh** [1], **Vethamonickam Stanley-Raja** [1], **Ramakrishnan Ramasubramanian** [1], **Aml Ghaith** [2], **Ahmed Abdel-Megeed** [3] and **Patcharin Krutmuang** [4,5,*]

1   Sri Paramakalyani Centre for Excellence in Environmental Sciences, Manonmaniam Sundaranar University, Alwarkurichi, Tirunelveli 627 412, Tamil-Nadu, India
2   Department of Zoology, Faculty of Science, Derna University, Derna 417230, Libya
3   Department of Plant Protection, Faculty of Agriculture Saba Basha, Alexandria University, Alexandria 21531, Egypt
4   Department of Entomology and Plant Pathology, Faculty of Agriculture, Chiang Mai University, Chiang Mai 50200, Thailand
5   Innovative Agriculture Research Center, Faculty of Agriculture, Chiang Mai University, Chiang Mai 50200, Thailand
*   Correspondence: senthil@msuniv.ac.in (S.S.-N.); patcharink26@gmail.com (P.K.)

**Abstract:** Among the most perilous factors affecting tomato plant functioning and yield is salinity. The efficacy of halotolerant marine macroalgal extract of *Chaetomorpha antennina* (Seaweed Extract—SWE) in mitigating the toxic effects of salt stress (150 mM) in tomato plants to promote and enhance both plant functionality and yield was tested. It was evident that salt stress undesirably affected germination and plant growth in terms of quality and quantity. Treatment with SWE improved the functionality of salt-strained tomato plants by enhancing their germination indexes, growth and morphological traits, and photosynthetic pigments, as well as protein and phenol concentrations. SWE also exerted a positive influence on protecting the plant against salt stress by increasing the synthesis and accumulation of antioxidant enzymes, superoxide dismutase and lipoxygenase enzymes, along with the contents of lycopene and vitamin C. SWE also increased the nutraceutical quality, flavour and organolepty of emerged tomato fruits. GCMS analysis of fruit pericarp showed increased siloxane, phenol, antioxidant and indole acetic acid compounds, along with aromatic benzene compounds. These results indicate the potentiality of SWE in protecting plants against salt stress induced toxicities by prompting the synthesis of protective compounds such as siloxane and antioxidant enzymes. It was also noted that SWE plays a crucial role in promoting plant growth and survivability by improving plant functionality, yield and nutrition, by promoting cultivation in saline soils in an eco-friendly and sustainable manner.

**Keywords:** seaweed extract; salt stress; tomato; alleviation; photosynthetic pigments; siloxane; antioxidant enzymes; vitamin C; fruit yield



## 1. Introduction

Tomato (*Solanum lycopersicum* L.) is one of the most popular vegetable species grown world-wide because of its edible fruit and agronomic importance. Tomato fruit is a crucial source of vitamins, minerals, and antioxidants for human wellbeing, and it is a key element of nutritional meals in various nations [1]. The bioactive compounds and health benefits of tomato fruits have recently received increased attention. According to a recent literature review, the most important characteristics of any vegetable product are nutrition and flavour [2]. According to data from Faostat, the world produced 186.821 million metric tonnes of tomatoes on 5,051,983 hectares in 2020, achieving an average yield of 37.1 metric

tonnes/hectare (mT/ha) [3]. Producing large yields of tomato fruits of superior quality and flavour is critical for ensuring consumer satisfaction.

The organoleptic quality of tomatoes is mostly determined by the quantity of sugar, in addition to acid. Their nutraceutical (nutritional) quality is primarily determined by their mineral, vitamin, flavonoid and carotenoid content [4]. Tomato flavour, on the other hand, differs, and depends on the cultivar and growing circumstances, and has been linked to greater amounts of reducing sugars and lower glutamic acid concentration. Carotenoids, instead, are a rich basis of vitamin A and antioxidizing agents, and so play a vital role in the deterrence of melanoma, as well as cardiac ailments [4]. Tomatoes are also high in flavonoids, which have anti-carcinogenic properties. Tomato fruit is also high in other bioactive chemicals including ferulic and caffeic acids, as well as low quantities of vitamins [5]. Fruits with high levels of bio-active secondary metabolites such as carotenoids and phenolics increase the nutritional qualities and chemical composition, resulting in a high added value yield with insect resistance [5,6]. As a result, improving fruit quality of plants subjected to salt stress is a critical subject for fresh foods with high nutritional content.

Salinity is an essential ecological stressor that decreases plant output. In addition to disrupting 20% of irrigated soil, salinization has been reported to limit crop productivity by one-third [7]. The process of salinization is influenced by changes in climatic trends. Reduced water availability in dry and semi-arid irrigated agricultural settings, salts rising from shallow water tables, the reuse of contaminated waters, and saltwater intrusion can all contribute to the development of soil salinity in the root zone [8]. Osmotic tension and ion toxicity are the two primary issues that plants face due to salinity. In salt-affected plants, nutritional problems and oxidative stress occur, resulting in a hazardous accumulation of sodium and chloride in the cytosol and cell organelle, as well as affecting water absorption [9]. Unsustained agriculture contributes to the increase in salinized land. Salt accumulation has affected the quality of a substantial expanse of the land around the world. The most efficient way to use such a big quantity of land is to yield salt-tolerant crop types [9]. Crop productivity and fruit quality may be negatively affected by saltwater in extreme salinity conditions. Appropriate salinity may well not be comprehensive since it depends on cultivar quality characteristics, interactions between cultivars, meteorological conditions, nutrient solution content and crop management [9].

To improve plant performance and provide protection against the deleterious effects of numerous abiotic stressors, a number of amendments such as biostimulators and bioelicitors have been used [10]. These compounds can be directed to germinating seeds or to plants throughout their vegetative development [10,11]. Seaweed extracts (SWE) are potential additives in this context for reducing the effects of salt stress on higher plant performance. Algae are well-known suppliers of plant macronutrients, micronutrients, and a number of key bioactive chemicals [11]. The abundant presence of growth-promoting components such as phytohormones, inorganic ions, amino acids, and vitamins makes SWEs a strong candidate for use in the improvement of both the organoleptic and nutraceutical traits of various crops [12]. The favourable impacts of using SWEs as natural regulators have resulted in better crop growth and production, along with the ability to withstand adverse environmental circumstances. The ability of SWEs to help plants alleviate abiotic stress is due to their enhanced radical scavenging activity. Salinity stress can induce nutrient deficiency by limiting their movement throughout the plant system, subsequently limiting their development. SWEs rich in micronutrients are reported to improve oxidative stress tolerance by promoting the synthesis of antioxidant enzymes, which not only helps to alleviate salt stress through their radical scavenging activity but also through enhancing micronutrient movement and distribution within the entire plant [11–13].

The relationship concerning the greenhouse atmosphere and tomato plant salinity is exceedingly complex. The relationship between physiological/biochemical backgrounds and tomato fruit consumer preference is also not well understood, and there are few detailed data on the relationship between tomato flavor and chemical composition when cultivated in a NaCl-enriched nutrient solution [9]. The objective of the current study

was to study the debarring effects of salt stress on the germination, growth, development and yield of tomato (PKM1). The constructive influence of SWE on the refinement of nourishment and the organoleptic characteristics of fruit was also studied.

## 2. Material and Methods

### 2.1. Seaweed Collection and Extraction

The seaweed *Chaetomorpha antennina* was collected from the rocks of seashore Colachel beach, Kanyakumari (8°14′5168″ N and 77°14′35.209″ E); December 2020), washed and processed. Extract of seaweed (SWE) was prepared by boiling processed seaweed pieces in sterile distilled water at 120 °C and 15psi pressure (100 g/L, for 1 h) [14].

### 2.2. Preparation of Treatment Solutions

A test solution of SWE (80%) was prepared by mixing SWE in sterile, distilled water. For the preparation of salt treatment solutions, NaCl (Merck, India) was used. Salinity treatment concentration (150 mM) was prepared by dissolving NaCl in distilled water (10 mL). Ammonium dihydrogen phosphate (Merck) was used as a starter fertilizer solution (SFS) by mixing 1 mg of $NH_4H_2PO_4$ in sterile, distilled water (10 mL), designated as a negative control for assays (NC) [15].

Commercially available seaweed-based biofertilizer was purchased and 10% of the solution was prepared by dissolving 1 mL of biostimulant in 10 mL of distilled water and was used as positive control (PC). Commercial SWE is a purified form and as per direction of usage the concentration made was 10%. Treatments involving salinity stress with NC and PC were labelled as SNC and SPC, respectively. Treatments involving sterile distilled water served as control (C). Since adverse effects of SWE and salinity were observed to be significant at the highest concentration, SWE 80%, Salt 150 mM and the combination of both were carried forward for all assays (S + SWE) (Table 1).

**Table 1.** Treatment solutions used in the study.

| T0 | T1 | T2 | T3 | T4 | T5 | T6 | T7 |
|----|----|----|----|----|----|----|----|
| Control | NC | NC + S150mM | PC | PC + S150mM | S150mM | SWE 80% | T5+ T6 |

where NC—negative control, S150 mM—salt at 150 mM concentration, PC—positive control, SWE 80%—seaweed extract at concentration 80%.

### 2.3. Tomato Seed Preparation

During the growing season, tomato seeds (PKM1) were purchased from Tamil Nadu Agricultural University stall (TNAU), Kadayam, South Tamil Nadu. Seeds were carefully selected for investigation, surface sterilized with 0.1 percent mercuric chloride, and rinsed three times in sterile distilled water [14].

### 2.4. Greenhouse Assay

Small pot field tests comprising sterilised soil were performed by transplanting 30-day tomato seedlings to examine the impacts of SWEs on tomato plant vegetative development under salt stress. The seedlings (2–3 true leaf stage) were transplanted into pot mixture (red soil: cow dung: vermiculate at 2:1:1, *w*/*w*/*w*) in surface sterilised (1% mercuric chloride) 15 cm diameter, 750 mL volume) pots at 1 seedling per pot. After 30 days of transplanting seedlings to appropriately labelled pots, they were irrigated with the respective treatment solutions (T0 to T7). Salinity was induced in the potting medium by adding 150 mM of salt in irrigation water 10 days after transplanting. For the next 2 weeks, salinity was maintained by irrigation with 150 mM salt water applied in 2-day intervals with the irrigation water. All of the treatments were irrigated twice a day with sterile, distilled water. Samples of vegetative development metrics were taken from seedlings and mature plants, chosen using a randomised block method [15].

### 2.5. Effect of Salinity and SWE on Tomato

2.5.1. Seed Germination

A filter paper test method was employed to test the effect of salt stress and SWE, separately and in combination, on the germination of tomato seeds (5 seeds/petri plate/treatment—5 replicates) by adding 2 mL of respective solutions. The effect was determined by estimating the germination percentage (GP) and index (GI) along with promptness (PI) as well as seedling vigour index (SVI). Germination stress tolerance index (GSTI) was also calculated [16,17].

$$GP = \frac{\text{Number of seeds germinated}}{\text{Total Number of seeds}} \times 100$$

$$GI = \frac{\sum G}{T}$$

where G is GP/day and T is the germination period.

$$PI = nd2\ (1) + nd4\ (0.8) + nd6\ (0.6) + nd8\ (0.4) + nd10\ (0.2)$$

where nd2, nd4, nd6, nd8, nd10 represent the percentage of germinated seeds after 2, 4, 6, 8 and 10 days.

$$SVI = \text{Seedling length (cm) germination \%}$$

$$\mathbf{GSTI} = \frac{\mathbf{PIS}}{\mathbf{PINS}}$$

where PIS—PI under salt stress and PINS—PI under normal conditions

2.5.2. Plant Growth

The mitigating effect of SWE on plant growth was determining the lengths of roots and shoots, along with determining the plant height (cm) [16]. Effect of salt as well the mitigating effect of SWE on the leaf membrane stability index (MSI), relative water content (RWC, %) and leaf area ($cm^2$) was also estimated [18].

MSI was calculated by measuring the electrical conductivity (EC) of leaf discs (7.5 mm diameter) cut from plants of all treatments, placed adaxial face down on 2 mL of sterilised ultrapure water in one well of a 12-well micro titre-plate, incubated for 30 min in a growth chamber (40 and 100 °C). EC was measured using an EC meter (Microprocessor EC Meter 1615, Parwanoo, India).

$$MSI = \frac{1 - C1}{C2} \times 100$$

where C1—EC at 40 °C and C2—EC at 100 °C.

Leaf relative water content (RWC) was determined from the fully expanded young leaves [18]. Fresh weight is the sample's fresh weight, dry-weight is the dry weight after oven-drying the leaves at 70 °C for 48 h, and saturated weight is the turgid weight after rehydrating the leaves at 4 °C.

$$RWC\ (\%) = \frac{\text{Fresh weight} - \text{Dry weight}}{\text{Saturated weight} - \text{Dry weight}} \times 100$$

Leaf area was calculated by grid method [19]. Leaves from each treatment were placed on 1 cm grid and their outlines were traced. The number of square centimeters covered, fully and partially, were counted and the resultant leaf area was expressed in $cm^2$.

2.5.3. Plant Functionality

The levels of antioxidant enzyme, superoxide dismutase (SOD) and lipoxygenase (LOX) were estimated as a measure of the plant's response to protect itself against salt stress. For SOD estimation, 1 g of frozen leaf samples from each treatment was ground in 50 mM sodium phosphate buffer, i.e., pH 7.8. The homogenates were centrifuged at 12,000× *g* for 20 min at 4 °C. The supernatants were used to measure the activity of the

enzyme spectrophotometrically at 440 nm, using a SOD determination kit (Sigma-Aldrich, INDIA) [20].

LOX levels were estimated by freezing 300 mg of the leaf samples from each treatment, homogenised in 50 mM sodium phosphate buffer (pH 7.0), 1 mM EDTA, 0.1 mM phenylmethylsulfonyl fluoride (PMSF), 2% (*w/v*) polyvinylpyrrolidone (PVP), 1% (*v/v*) glycerol, and 0.1% (*v/v*) Tween 20. The extract was centrifuged at 15,000× *g* for 20 min and the supernatant was immediately used to assay for LOX activity by measuring it spectroscopically at room temperature by adding 1 mM linoleic acid in 0.1 M sodium acetate buffer (pH 5.6) to the extract and reading the increase in absorbance at 234 nm [21].

The salt stress alleviating effect of SWE on plant functionality was also determined by estimating their protein and phenol contents, by using the enzyme extract prepared for SOD estimation [15].

### 2.5.4. Nutraceutics

The effect of SWE on the nutraceutical quality of salt stressed tomato fruit was estimated by analysing the levels of photosynthetic pigments (chl a and b), lycopene, and carotenoids [22,23]. Slurry of tomato fruits from each treatment (1 g) was homogenised with 10 mL of an acetone–hexane mixture (2:3) for 2 min to uniform mass, centrifuged at 5000 rpm for 10 min at 20 °C, and absorbance was measured at 453, 505, 645 and 663 nm. Parameters were calculated based on the following formulae.

$$\text{Chl a} = 0.999A_{663} - 0.989A_{645}$$

$$\text{Chl b} = -0.328A_{663} + 1.77A_{645}$$

$$\text{Lycopene} = -0.0485A_{663} + 0.204A_{645} + 0.372A_{505} - 0.0806A_{453}$$

$$\text{Carotenoids} = 0.216A_{663} - 1.22A_{645} - 0.304A_{505} + 0.452A_{453}$$

Levels of soluble sugars (%) were estimated using the phenol-sulphuric acid method, and the pH of the tomato juice was also determined [17,18].

### 2.5.5. Fruit Organolepty

The effect of SWE on the fruit organoleptic traits was found by estimating their firmness (kg/cm$^2$), measured by penetrometer (lab_35459, Labpro, India) [24], fresh weight (g/fruit), and visually interpreting their colours [25].

### 2.5.6. Metabolomics—GCMS

GCMS was performed to analyse the composition differences in the pericarp of tomato fruit brought about by salt stress and treatment with SWE. Tomato fruits (2/test) were collected (45 days after flowering, red ripe stage) and washed with sterile, distilled water. The tomatoes were ground using liquid nitrogen (4.3 g), added to $CaCl_2$ (1.7 mL, saturated) in glass vials, which were closed and sealed immediately, and carried forward for GCMS analysis using a GCMS unit (Agilent Technologies, 7820A), 5977E MSD, column -DB-5. The sample was injected in a 1:2 split mode using helium as the carrier gas at 3 mL min−1 with a temperature of 100–270 °C (10 °C /min). Compound identification was carried out using MS library [26].

### 2.6. Statistical Analysis

All of the tests were carried out five times in total. Using the Minitab®17 software programmer, the effect of SWE and salt stress on tomato plants was investigated using analysis of variance, one-way (ANOVA), and the treatment means were compared using the Tukey-family error test ($p < 0.05$). Sigmaplot 11 was used to create the graphs.

## 3. Results

### 3.1. Germination Parameters

Salt stress severely affected the germination of treated seeds, reducing the GP by 69.32% compared to the control ($F_{7,32}$ = 290.93; $p < 0.0001$) (Figure 1). The GI of salt treated as well as the control seeds were not found to be significantly different ($p > 0.05$). However, the PI of salt stressed seeds was reduced to 65.56% compared to the control ($F_{7,32}$ = 36.1; $p < 0.0001$). SWE treatments significantly induced the germination parameters of seeds under salinity stress increasing the GP to 80.6 from 20% ($F_{7,32}$ = 290.93; $p < 0.0001$). A conforming upsurge in GI and PI of SWE treated seeds in salt stress by 83.04 and 83.27% were also observed (Figure 1B,C). SWE also drastically improved the GSTI of salt-stressed seeds by 83.64% ($F_{7,32}$ = 358.46; $p < 0.0001$) (Figure 1D).

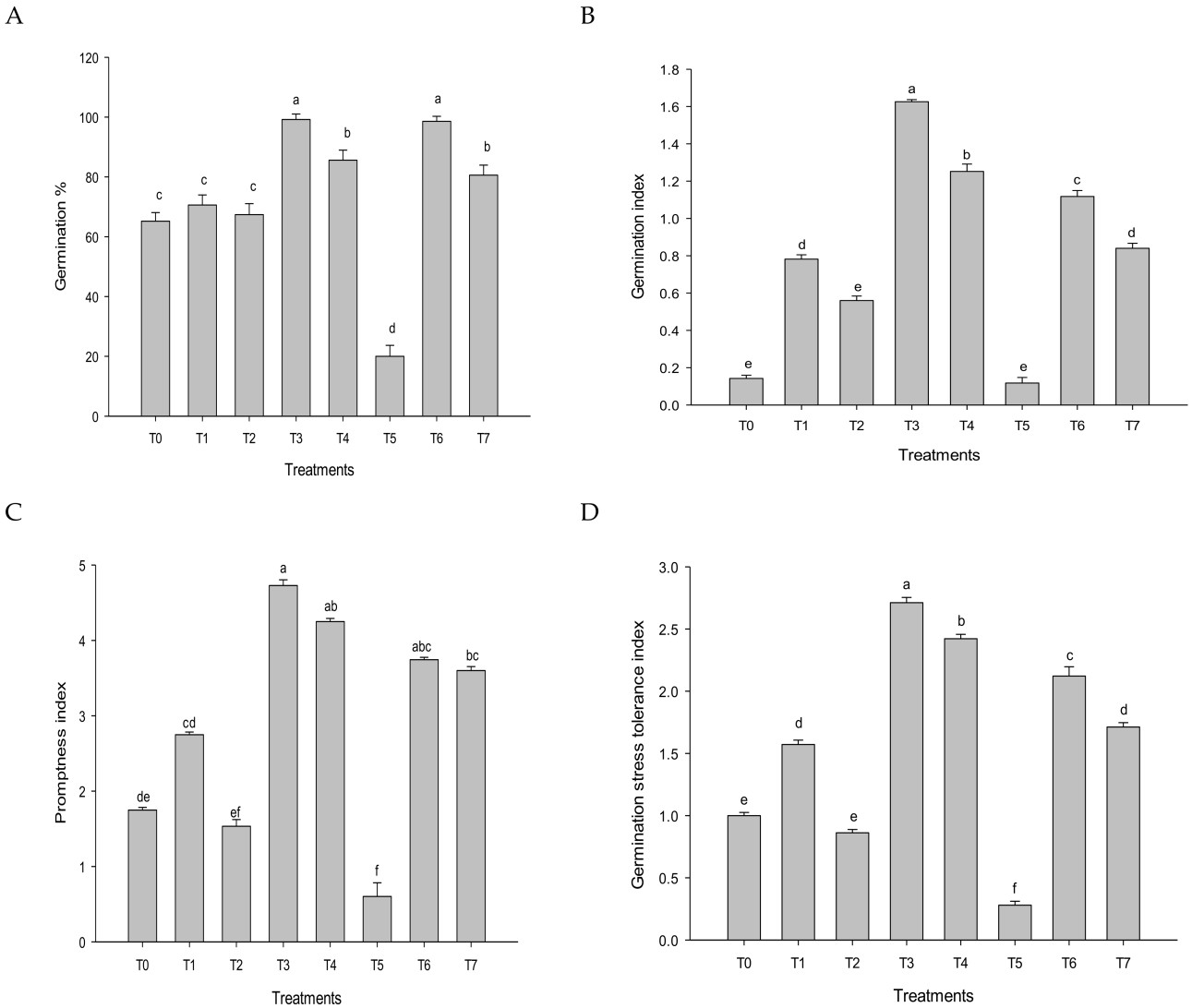

**Figure 1.** Effect of SWE on mitigating seed germination capabilities of salt-strained tomato seeds; (**A**) Germination Percentage; (**B**) Germination Index; (**C**) Promptness Index; (**D**) Germination Stress Tolerance Index. Mean (± SEM) of 5 replicates. Within each time point, the same letter indicates no significant difference ($p \leq 0.05$) in a Tukey's test. (T0—Control, T1—NC, T2—NC + S150 mM, T3—PC, T4—PC + S150 mM, T5—S150mM, T6—SWE 80%, T7—T5 + T6).

SWEs enhanced the vigour of salt-stressed seedlings by 94.136% as a consequence of enhancing the length of salt-stressed seedlings from 2.6 to 8.52 cm ($F_{7,32}$ = 634.68; $p < 0.0001$) (Table 2, Figure 2).

**Table 2.** Effect of SWE on mitigating salt stress on SVI, RWC(%) and leaf area (cm$^2$).

| Treatments | SVI | RWC (%) | Leaf Area (cm$^2$) |
|:---:|:---:|:---:|:---:|
| $T_0$ | 254.66 ± 3.97 [g] | 72.61 ± 0.0498 [g] | 8.44 ± 0.1817 [b] |
| $T_1$ | 903.74 ± 2.59 [d] | 79.93 ± 0.302 [e] | 9.2 ± 0.2121 [ab] |
| $T_2$ | 504.9 ± 3.51 [f] | 75.37 ± 0.038 [f] | 8.84 ± 0.1817 [ab] |
| $T_3$ | 1506.97 ± 5.09 [b] | 93.66 ± 0.0522 [a] | 10.2 ± 0.2121 [a] |
| $T_4$ | 1052.78 ± 3.96 [c] | 86.41 ± 0.0474 [c] | 9.72 ± 0.228 [ab] |
| $T_5$ | 40.2 ± 5.07 [h] | 55.46 ± 0.0507 [h] | 4.6 ± 2.074 [c] |
| $T_6$ | 1596.66 ± 5.23 [a] | 89.9 ± 0.264 [b] | 9.88 ± 0.239 [ab] |
| $T_7$ | 685.62 ± 3.97 [e] | 82.69 ± 0.044 [d] | 9.04 ± 0.416 [ab] |

Columns denoted by a different letter are significantly different at $p \leq 0.05$ when estimating the differences among treatments on the SVI and leaf RWC and area. (T0—Control, T1—NC, T2—NC + S150 mM, T3—PC, T4—PC + S150 mM, T5—S150 mM, T6—SWE 80%, T7—T5 + T6).

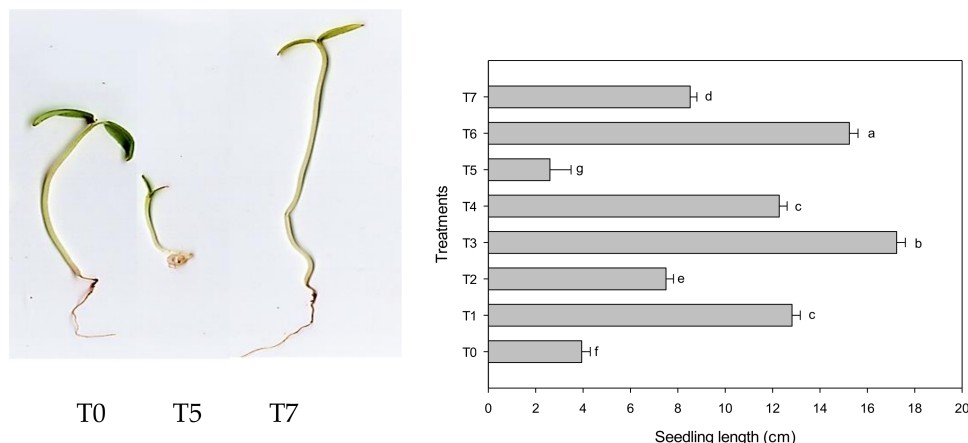

**Figure 2.** Effect of SWE on mitigating seedling length of salt-strained tomato seeds. Mean (± SEM) of 5 replicates. Within each time point, the same letter indicates no significant difference ($p \leq 0.05$) in a Tukey's test. (T0—Control, T1—NC, T2—NC + S150 mM, T3—PC, T4—PC + S150 mM, T5—S150 mM, T6—SWE 80%, T7—T5 + T6).

### 3.2. Growth Parameters

Salt stress had a negative impact on the development factors of tomato plants, reducing the length of root to 1.52 from 4.38 cm (Figure 3). However, SWEs enhanced the lengths of roots to 9.06 cm, increasing their growth by 83.22% ($F_{7,32} = 74.21$; $p < 0.0001$) (Figure 3). A consistent intensification in the lengths of the shoots as well as plant height was also observed, which rose from 2.66 and 5.3 cm to 9.06 ($F_{7,32} = 132.99$; $p < 0.0001$) and 20.06 cm ($F_{7,32} = 206.13$; $p < 0.0001$) (Figure 3). The SWE promoted plant growth by 73.57%.

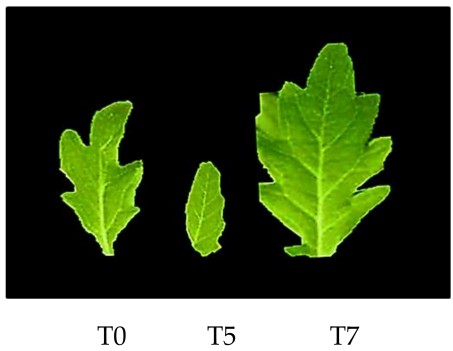

**Figure 3.** Effect of SWE on mitigating salt stress effects on leaf surface area expansion. (T0—Control, T5—S150 mM, T7—T5 + T6).

Leaf area was also considerably reduced by salt stress (Table 2). A large increase in the area of leaf was caused by the action of SWEs to 9.04 from 4.6 cm$^2$ ($F_{7,32}$ = 26.45; $p < 0.0001$) (Figure 3).

### 3.3. Physiological Parameters

Salt stress debarred the MSI of tomato leaves, reducing stability by 4.5% (Figure 4). The MSI of salt-affected leaves were re-stabilised using SWE treatment by 51.40% ($F_{7,32}$ = 170.41; $p < 0.0001$). A similar reduction in the RWC of salt-stressed leaves was observed that was enhanced by SWE treatment from 55.46 to 82.69% (Table 2).

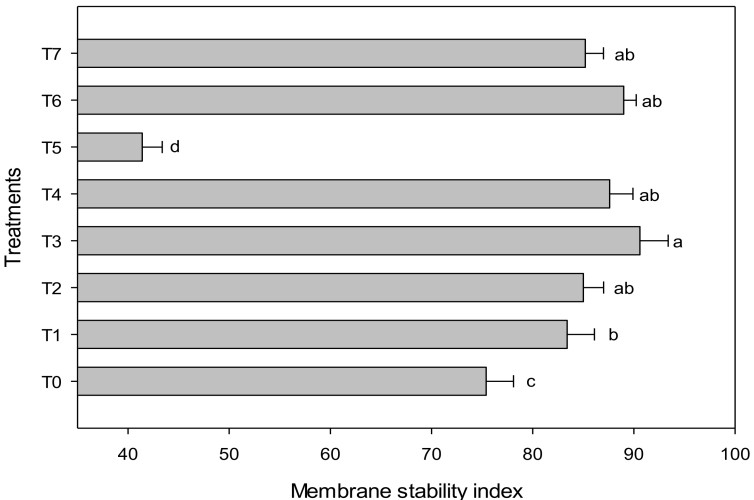

**Figure 4.** Effect of SWE on mitigating salt stress effects on the membrane stability index of tomato leaves. Mean (±SEM) of 5 replicates. Within each time point, the same letter indicates no significant difference ($p \leq 0.05$) in a Tukey's test. (T0—Control, T1—NC, T2—NC + S150mM, T3—PC, T4—PC + S150 mM, T5—S150 mM, T6—SWE 80%, T7—T5 + T6).

### 3.4. Plant Functionality

SWE enhanced the functionality of salt-stressed plants by positively influencing the synthesis and accumulation of plant protein, phenols, antioxidant enzyme SOD as well as LOX enzyme levels (Figure 5A). The protein content of salt-stressed leaves decreased to 1.018 mg/gFW was increased to 1.56 mg/gFW ($F_{7,32}$ = 63.1; $p < 0.0001$). An increase in the phenol content of leaves of 40.06% ($F_{7,32}$ = 72.14; $p < 0.0001$) (Figure 5B) was also observed. Both SOD and LOX levels decreased as a result of salt stress to 1.14 and 0.78 from 1.56 and 1.4, increased by 41.11 ($F_{7,32}$ = 52.36; $p < 0.0001$) and 59.45% ($F_{7,32}$ = 57.21; $p < 0.0001$), respectively.

A

B

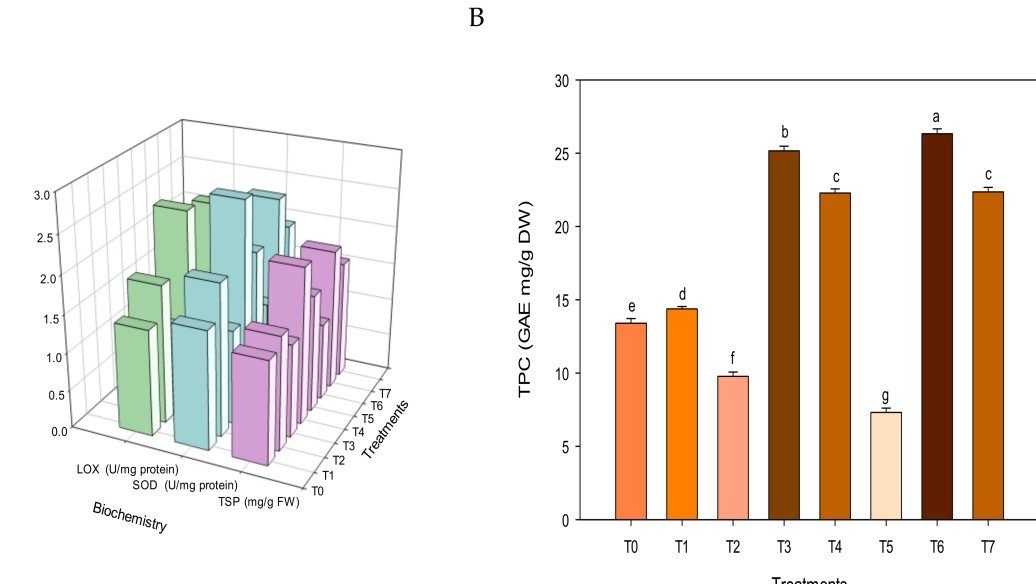

**Figure 5.** Effect of SWE on mitigating salt stress effects on plant functional metabolites (**A**) TSP (mg/mg FW); SOD and LOX (μg/mg protein); (**B**) Total phenol contents (GAE mg/g DW). Mean (± SEM) of 5 replicates. Within each time point, the same letter indicates no significant difference ($p \leq 0.05$) in a Tukey's test. (T0—Control, T1—NC, T2—NC + S150 mM, T3—PC, T4—PC + S150 mM, T5—S150 mM, T6—SWE 80%, T7—T5 + T6).

### 3.5. Nutritional Parameters

The salt stress severely affected the photosynthetic pigments, *chl a*& *b* and carotenoid levels, drastically reducing their concentrations to 6.1, 1.8 and 10.92 mg/g FW (Figure 6A). However, the pigment levels were increased by SWE treatment to 10.58 ($F_{7,32} = 18.37$; $p < 0.0001$), 5.16 ($F_{7,32} = 286.37$; $p < 0.0001$) and 13.32 mg/gFW ($F_{7,32} = 82.24$; $p < 0.0001$). A 42.34, 5.15 and 18.04% increase in photosynthetic pigments was brought about by SWE treatment on salt-stressed tomato plants.

A

B

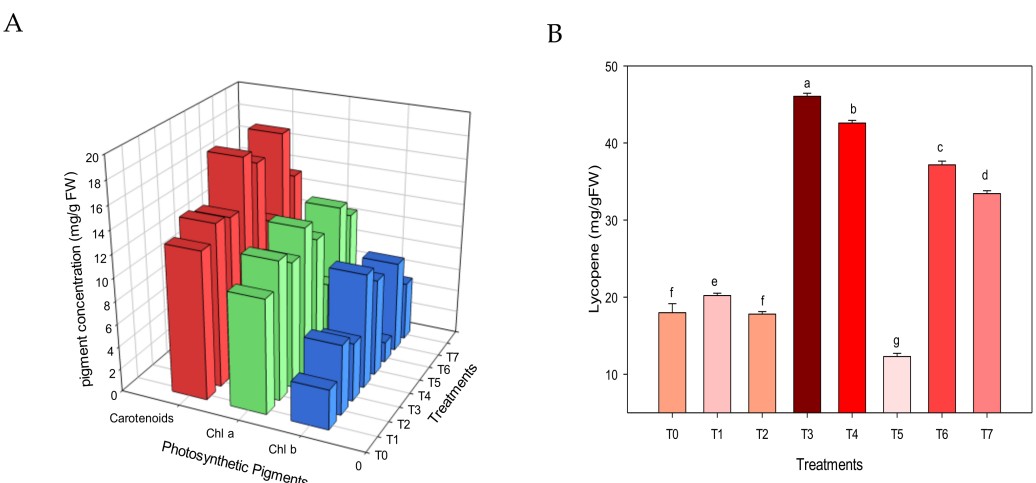

**Figure 6.** Effect of SWE on mitigating salt stress effects on fruit pigments (**A**) Photosynthetic pigments (Chlorophyll a, b and carotenoids—mg/g FW); (**B**) Lycopene (mg/g FW). Mean (±SEM) of 5 replicates. Within each time point, the same letter indicates no significant difference ($p \leq 0.05$) in a Tukey's test. (T0—Control, T1—NC, T2—NC + S150 mM, T3—PC, T4—PC + S150 mM, T5—S150 mM, T6—SWE 80%, T7—T5 + T6).

An analogous escalation in the lycopene level that was reduced from 17.98 to 12.3 mg/g FW by salt stress was increased to 33.44 mg/g FW, resulting in a 63.21% increase ($F_{7,32} = 27.04$; $p < 0.0001$) (Figure 6B).

Conversely, the pH of tomato juice was not severely affected by salt stress (Table 3). The pH of fruit juice from salt-affected tomato plants was observed to be 3.78 and that of control was 4.04. SWE treated salt-stressed plants produced fruits with juice of pH 4.07 ($F_{7,32} = 2.22$; $p < 0.059$). Vitamin C levels and pigment concentrations of fruit affected by salt stress were enhanced by SWE (57.73%) (Figures 7 and 8).

**Table 3.** GCMS analysis tomato pericarp in treatments T0, T5 and T7 (T0—Control, T5—S150 mM, T7—T5 + T6).

| Compounds | Peak Area % | | |
|---|---|---|---|
| | **T0** | **T5** | **T7** |
| Antioxidants | | | |
| 2-Ethylacridine | 20.18 | 4.34 | 8.38 |
| 4H-Pyran-4-one,2,3-dihydro-3,5- | – | 4.27 | 26.37 |
| Alkanes | | | |
| Cyclotrisiloxane, hexamethyl | 8.98 | 5.27 | 10.82 |
| Octasiloxane | 70.84 | 9.34 | 12.71 |
| Methyltris(trimethylsiloxy)silane | 5.88 | 7.71 | 16.52 |
| Aromatic benzene compounds | | | |
| 1,2-Bis(trimethylsilyl)benzene | – | 3.03 | 10.82 |
| 1,2-Benzisothiazol-3-amine tbdms | 20.18 | 6.62 | 19.76 |
| Fatty acid | | | |
| Benzo [h] quinoline, 2,4-dimethyl | – | 14.06 | – |
| Phenols | | | |
| Ethylene glycol phenyl ether methacrylate | – | – | 17.91 |
| Indole acetic acid derivative | | | |
| 1H-Indole, 1-methyl-2-phenyl- | – | 1.45 | 4.06 |

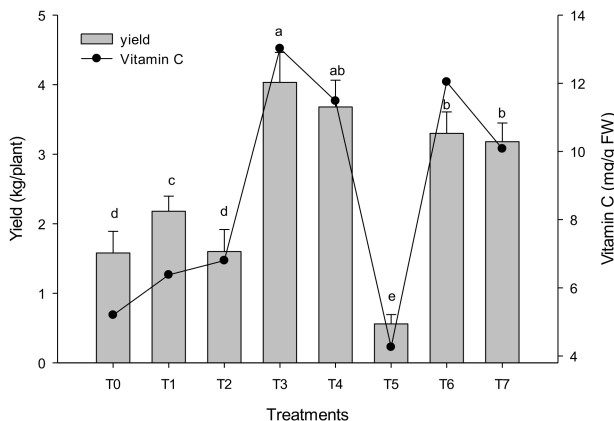

**Figure 7.** Effect of SWE on mitigating salt stress effects on fruit yield (kg/plant) and Vitamin C (mg/g FW). Mean (±SEM) of 5 replicates. Within each time point, the same letter indicates no significant difference ($p \leq 0.05$) in a Tukey's test. (T0—Control, T1—NC, T2—NC + S150 mM, T3—PC, T4—PC + S150 mM, T5—S150 mM, T6—SWE 80%, T7—T5 + T6).

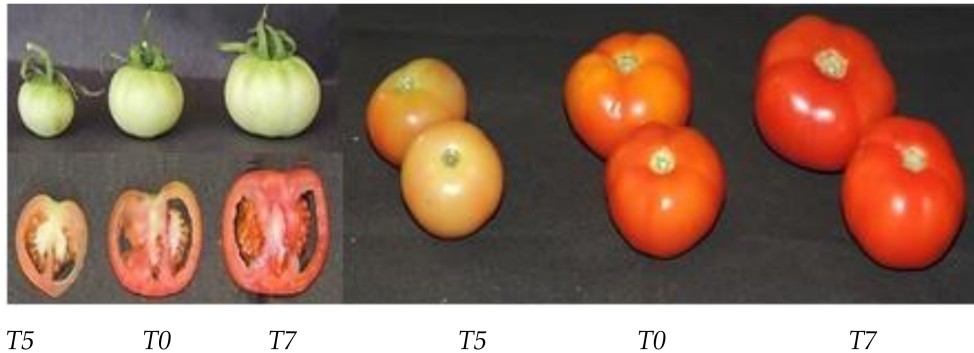

| T5 | T0 | T7 | T5 | T0 | T7 |

**Figure 8.** Effect of SWE on mitigating salt stress effects on fruit organolepty. (T0—Control, T5—S150 mM, T7—T5 + T6).

### 3.6. Fruit Organolepty

Fruit qualities such as steadfastness and fresh weight were also affected by salt stress (Table 3). Firmness that was reduced to 1.72 from 2.216 $kg/cm^2$ was increased by SWE (3.14 $kg/cm^2$) ($F_{7,32}$ = 116.59; $p < 0.0001$) (Table 4). A 29.58% increase in fruit firmness was brought about by SWE treatment on salt-stressed fruit-bearing plants. The yield of tomato plants under salt stress was positively influenced by SWE treatment (Figure 7). SWE increased the yield of tomato plants in salt stress from 0.86 to 3.18 kg/plant, which was 72.95% higher than the yield of salt-stressed plants ($F_{7,32}$ = 91.66; $p < 0.0001$). The fresh weight of tomatoes as a result of SWE-treated salt-stressed plants showed a considerable increase (Table 3). The fresh weight of salt-stressed tomato fruit was 15 g, which was increased by SWE to 25.06 g ($F_{7,32}$ = 85.29; $p < 0.0001$). SWE was successful in raising the fruit fresh weight of salt-stressed tomato plants by 40.14%. The colour of tomato fruits as recorded by visual interpretation was pale red to mild, and dark red in fruits of plants under treatments T5, T0 and T7, respectively. This could be correlated with the higher levels of pigment concentration in T7 compared to T5 (Figure 8).

**Table 4.** Effect of SWE on mitigating salt stress on fruit juice pH, fruit firmness ($kg/cm^2$) and fruit fresh weight (g/fruit). (T0—Control, T1—NC, T2—NC + S150 mM, T3—PC, T4—PC + S150 mM, T5—S150 mM, T6—SWE 80%, T7—T5 + T6).

| Treatments | pH | Firmness (kg/cm²) | Fruit Fresh Weight (g/fruit) |
|---|---|---|---|
| T0 | 4.04 ± 0.1537 [ab] | 2.21 ± 0.1228 [c] | 19.94 ± 0.305 [g] |
| T1 | 4.20 ± 0.255 [ab] | 3.08 ± 0.0377 [d] | 24.04 ± 0.365 [e] |
| T2 | 4.04 ± 0.261 [ab] | 2.30 ± 0.1188 [d] | 22 ± 0.474 [f] |
| T3 | 4.37 ± 0.432 [a] | 4.46 ± 0.321 [a] | 32 ± 0.316 [a] |
| T4 | 4.21 ± 0.2151 [ab] | 4.17 ± 0.2046 [a] | 29.64 ± 0.78 [b] |
| T5 | 3.78 ± 0.228 [b] | 1.72 ± 0.327 [e] | 15 ± 0.316 [h] |
| T6 | 4.24 ± 0.364 [ab] | 3.67 ± 0444 [b] | 28.04 ± 0.365 [c] |
| T7 | 4.07 ± 0.0825 [ab] | 3.17 ± 0.2046 [c] | 25.06 ± 0.397 [d] |

Columns denoted by a different letter are significantly different at $p \leq 0.05$.

### 3.7. GCMS

The pericarp GCMS analysis showed the presence of 11 compounds in salt stressed pericarp of tomato fruit, whereas 12 compounds were seen to be present in the control and in seaweed-extract-treated salt-stressed tomato plants (GCMS Spectrum –attached as Supplementary Material).

Alkanes were found to be in higher concentration in the pericarp of SWE-treated tomato plants. The concentration of benzene aromatic compounds was also increased by SWE treatments. In the pericarp of salt-stressed tomato fruits, higher levels of indole acetic compounds were detected (Table 4).

## 4. Discussion

Salinity stress is a most unfavourable abiotic cause of crop development and yield loss. Several management strategies have been used to lessen the detrimental effects by mediating either fast clearance of toxic ions from the soil solution or their sequestration into less sensitive organelles, while simultaneously boosting existing tolerance mechanisms [7,8]. In this regard, the introduction of innovative mitigating chemicals can boost plant development and production in salty environments. The usefulness of a liquid extract of the marine green macroalgae, *Chaetomorpha antennina*, in alleviating the unfavourable effects of salt on tomato plants was investigated in this study. While salt stress has been reported to have various damaging effects on plants all through their growth, the mitigating effect of SWE on tomato plants from seed germination to plant growth was investigated by estimating the germination, growth, metabolite accumulation and fruit traits.

Stress-free growth conditions at the germination stage are a crucial determinant of plant development and production. In this study, tomato seeds were shown to propagate under different treatment conditions of salt and SWE, both individually and in combination. Salinity severely affected the germination capabilities of tomato fruit, lowering their GP by 69.32%, which eventually reduced their PI (65.56%), producing seedlings of stunted growth and a very poor SVI. Salt-stressed seeds were clearly observed with a very lower GSTI. A probable effect of the salinity strain on reduced GP, PI, SVI and GSTI was reported earlier by Tanveer et al. [27], who studied the effects of salt strain on the germination capabilities of tomato seeds. Seedling growth inhibition was reported as an early response to salt stress [28]. SWE increased the overall germination competencies of salt-stressed tomato seeds, resulting in an increased germination and a promptness index, which are early indicators, validating the emergence of plants that are consistently salt-tolerant. The SWE-primed seeds were observed with higher GSTI and PI, indicating that the extracts not only conferred protection to salt stress but also prompted early germination.

A likely enhancement in the stress tolerance index of seeds primed with seaweed extract in *Calotropis procera* was reported by Bahmani et al. [29]. This beneficial impact of algal extract is ascribed to the manifestation of various salt-ameliorating bio-active molecules encompassing amino acids and phytohormones along with pest control properties [30]. The EC of the environment is considered to be an important factor for promoting seed germination along with oxygen availability. Seaweed extracts are reported to possess an optimum EC that is favourable for seed germination and development, and are also known to increase oxygen availability to the embryo. These traits have promoted germination of tomato seeds treated with seaweed extracts [14,16].

Plants modify the extracellular pH as a necessary adaptation to salt stress, which impairs the roots' capacity to tolerate salt stress [31]. Failure to do so has an impact on root growth, reducing the plant's ability to establish itself. Although salinity stress resulted in seedlings of poor vigour, reducing their length, SWE augmented the vigour of tomato seedlings by 69.4%. Extracts of *C. antennina* were reported with the presence of several plant essential micro as well as macro elements, which have the potential to neutralise the pH and reduce water stress by enhancing the imbibition capabilities of primed seeds, thereby promoting cell division in roots [32]. Salinity stress also significantly affected numerous plant development factors of tomato plants, producing plants of stunted growth, with correspondingly lower root-shoot heights and leaf area, essentially as a result of degenerative cell division process. However, SWE treatment boosted the growth parameters of plants (54.83%), along with leaf expansion (49.1%). A similar mitigating effect of glycine betaine, a major component of seaweed, on the development of salt-stressed *Dalbergia odorifera* was reported by Cisse et al. [32]. The growth-promoting effect could be correlated with the presence of high amounts of plant hormones, for instance auxins, cytokinins and gibberilic acid, in seaweed extracts [33].

The membrane stability index (MSI) and relative water content (RWC) of the leaves were severely affected by salt stress. Higher osmotic potential instigated by salt exposure disrupts membrane stability of plant cells, affecting ion homeostasis, causing ion

toxicity [34]. Salinity has furthermore been testified to limit water movement within the system, deliberately inducing physiological drought [35]. These undesirable effects were removed using SWE application, which not only re-stabilised the membranes, reducing ion leakage, but also increased the water retention capabilities of the salt-stressed plants, due to the ability of seaweed extract to increase the water-withholding capability of salt-stressed alfalfa [36]. This was achieved by promoting nutrient uptake and movement within the plant system, influenced by the presence of microelements such as Ca and Mg [27]. Seaweed extracts are known to possess osmolytes such as mannitol, which is an essential abiotic stress protectant that boosts root development and soil water-holding capacity in salt-stressed plants [37].

Reactive oxygen species are formed as a result of salt stress. The main detrimental consequences of ROS under salt stress are loss of membrane reliability, oxidation of carbohydrates, and nucleic acid oxidation. Under stress, ROS levels rise, the equilibrium between antioxidant defence mechanisms deteriorates, and oxidative stress develops [38]. In this context, enzymes such as LOX and SOD, protectors of plant functionality against abiotic stress, are severely de-regulated by high levels of salinity, resulting in plants with much lower protein and phenol contents. However, SWE profoundly increased the activities of LOX and SOD, consequently enhancing the plant's protein and phenol contents. Overproduced ROS, which results in lipid-peroxidation-induced apoptosis, was reversed by SWE, which was reported to contain higher amounts of antioxidants as well as phenolic compounds. Reduced ROS fabrication in seaweed *Kappaphycus alvarezii* sap-treated *Triticum durum* plants exposed to both salinity and drought resulted in increased concentrations of non-enzymatic antioxidants such as total phenols and an increased expression of SOD and catalase genes, as previously reported [39].

Salinity has a long-term and short-term impact on photosynthesis. The most significant effects of salt are thylakoid membrane deterioration and a lessening in Calvin cycle enzyme activity. Salinity-induced decreases in photosynthetic pigment deliberation might be a result of increased pigment breakdown or decreased production [40]. A direct correlation between leaf area and photosynthetic pigment concentration was also observed. SWE were found to exert a positive influence on the contents of photosynthetic pigments, increasing the concentrations of chlorophyll (a and b), along with carotenoids. Marine macroalgal extracts have been shown to intensify the chlorophyll content of treated plants such as mung bean, garden cress, and wheat in several experiments [40,41]. The existence of betaines in marine macroalgal extracts may be accountable for the increase in chlorophyll level in plants following treatment [36]. The increase in total chlorophyll content was linked to a higher net photosynthetic rate induced by SWE, as well as an increase in the intensities of carotenoid, which might be one of the causes that helped to relieve salt stress [42,43]. Increased carotenoids production as a result of SWE treatment may have aided photosynthetic fortification by arbitrating ROS foraging and contributing to redox stability [44].

The positive effect of SWE on fruit quality and organoleptic parameters such as lycopene vitamin C contents, as well as firmness, pH and TSS, respectively, were observed. Firmness is an essential sign of tomato fruit progress and a criterion of tomato fruit eminence influenced by cell wall integrity besides turgidity. Fruit firmness increased in plants exposed with SWE and grew under salt stress. The upright effect of SWE on fruit firmness may be explained by their impacts on ethylene generation and potassium content; algae extract also has a positive influence on turgidity and cell wall components, which may improve cell membrane flexibility and fruit firmness [45]. Lycopene, a red carotene-based pigment, is linked to tomato fruit quality [46]. The rise in carotenoid levels resulting from SWE treatment might be directly connected to an increase in lycopene concentration, resulting in high-quality fruits. TSS is a major component of tomato fruit which is crucial for determining quality of tomato, influencing the strength of their flavour [47]. The use of natural polysaccharides in tomato fruit has already been shown to improve oragnolepty and nutritional characteristics. The levels of sulphate groups and carbohydrates in SWE

can also explain enhanced development and production, as well as fruit eminence in SWE-treated plants. The content of ascorbic acid is highly dependent on environmental factors. Apart from being involved in plant adaptation pathways during stress circumstances and functioning as antioxidants, ascorbic acid levels impact the nutritional content of tomato fruit [27].

While pericarp tissue metabolome could be used as a direct quotient to determine the value of tomato fruits, a GCMS analysis of pericarp tissue of tomato fruits developed under salt stress was carried out. The results implied the presence of higher amounts of siloxanes in the fruit pericarp tissue of SWE-treated salt-stressed tomato plants. Salt stress significantly lowered the level of siloxanes, which was evident from the lower levels of occurrence in tomato fruits which resulted in salt-stressed tomato plants. In a previous study, the negative consequence of salt stress on alkane concentration of tomato leaves and the positive influence of microalgal extract in increasing the siloxane concentration was reported [48].

As a physiological response to salt stress, tomato plants secrete antioxidant enzymes to mitigate salinity-induced excess ROS produced within the plant system [49]. This was evident from the existence of anti-oxidant compounds in salt-stressed tomato fruits that was magnified further by the action of SWE. A corresponding increasing in the blend of phenol compound and aromatic benzene compounds, along with indole acetic acid compounds, were also detected in fruits of SWE-treated salt-stressed tomato plants. In a previous study, the potential of bacterial colonisation to induce secretion of antioxidant enzymes such as SOD and antioxidant compounds to scavenge excess ROS was reported, which helped alleviate salt stress effects in tomato. They also reported the accumulation of indole acetic acid compounds in tomato leaves that attributed to enhanced development strictures of tomato plants exposed to salt stress, similar to our study [50]. Fatty acid presence in fruits of salt stressed tomato plants was detected, which was absent in both the control and in SWE-treated tomato fruits. The presence of the long chain fatty acid is evidence that the plant has been subjected to salt stress [51]. Phenol compound and aromatic benzene compounds presence was also detected in higher levels in SWE-treated fruits of salt-stressed tomato plants. Salt stress reduces plant growth and development, alters carbon metabolism and nutritional status, and oxidative metabolism, and changes secondary metabolite levels such as phenols and aromatic benzene compounds, which are key physiological markers in salt stress resistance [51].

## 5. Conclusions

Salt stress reduces tomato plant growth and impairs physiological function, lowering crop production. Germination assays revealed that SWE were successful in inducing early germination and promoting the establishment efficiency of the seedling. SWE were also successful in enhancing the germination stress tolerance index of treated seeds. Our findings from the greenhouse experiment revealed that using SWE would benefit the development tomato plants exposed to salt stress, which was evident from the observations made on plant vegetative growth (plant height and leaf area) and functional traits (enhanced oxidative enzyme secretion, as well as increased accumulations of protein and phenol). Fruit yield, weight and nutritional as well as organoleptic traits such as firmness, pigment concentration, TSS and pH of juice were also enhanced by SWE by promoting the adaptation mechanisms of tomato plants growing under salt stress by accumulating metabolites. As a result of the activation of antioxidant enzymes and synthesis of osmoprotectants, SWE treatment resulted in an ameliorative reaction. SWE application also enhanced the synthesis of siloxanes that was detected through GCMS in tomato fruit pericarp, which is responsible for fruit flavour. This study suggests that using SWE to defend tomato plants from salt stress is a sustainable as well as eco-friendly agronomic strategy.

**Supplementary Materials:** The following supporting information can be downloaded at: https://www.mdpi.com/article/10.3390/agriculture13010006/s1, Figure S1: GCMS Spectrum T0, T5 and T7 pericarp; Figure S2: Compounds of T0, T5 and T7 pericarp.

**Author Contributions:** Conceptualization—K.M.-P.C. and S.S.-N.; Methodology—K.M.-P.C. and G.-S.P.; Validation—K.M.-P.C. and S.S.-N.; Investigation—G.-S.P., A.-S.A., P.M. (Pauldurai Malarvizhi), V.S.-R., A.G. and P.M. (Ponnusamy Murugan); Formal analysis—A.D.-A., P.M. (Pauldurai Malarvizhi), H.S., V.S.-R., and R.R.; Data curation—K.M.-P.C., S.S.-N.; writing—original draft preparation, K.M.-P.C.; writing—review and editing, A.A.-M., P.K. and S.S.-N. All authors have read and agreed to the published version of the manuscript.

**Funding:** This research was supported by the department of science and technology (DST-FIST), India under FIST program (SR/FIST/LS-1/2019/522). This research was also partially funded by Chiang Mai University, Thailand.

**Institutional Review Board Statement:** Not Applicable.

**Informed Consent Statement:** Not Applicable.

**Data Availability Statement:** Data are contained within the article.

**Conflicts of Interest:** The authors declare no conflict of interest.

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
