# Peer review of "The Macroalgal Biostimulant Improves the Functional Quality of Tomato Fruits Produced from Plants Grown under Salt Stress"

_agriculture, doi:10.3390/agriculture13010006_

Round 1

Reviewer 1 Report

This publication presents interesting results related to the alleviation of salt stress on tomatoes by using SWE, through promoting growth, physiological, biochemical, and fruit quality traits. The adopted approach in this study was very interesting since it could a sustainable approach to mitigate the salinity stress on plants.

The manuscript was well introduced, and the authors adopted convincing methods with a discussion of the different obtained results. However, the manuscript needs substantial revisions to be suitable for publication in Agriculture.

General comments

- Comment 1: The English of this manuscript needs substantial improvements with a check of the punctuation.

- Comment 2: Many choices made in M&M should be justified and many protocols are not well described.

- Comment 3: in the results section, please use % of variation instead of the recorded values.

Comment 4: The discussion must be more developed including mechanisms to explain the recorded changes in the assessed parameters.

Other comments

Abstract

- L24: please add a space between the value and the unit. Please correct throughout the manuscript

- L30: please change “nutritional” to “nutraceutical”.

- Keywords: please change “marine algae” to “seaweed extract” and add salinity and tomato.

Introduction

- L45: at the beginning of the sentence you stated “According to recent studies” but you mention just one reference. Please correct.

- L57-58: please change “carotenoids, phenolics” to “carotenoids and phenolics”.

- L94: please change “SE” to “SWE”.

- L94-95: please rephrase.

M&M

- L98: please write “Materials” without “s”

- L101: please delete figure 1 and just put  the GPS coordinates of the site.

- L103: please correct “((100 g/1 L- 1 h)”

- L107: Is it “150 ppm” or “150 mM”? please correct.

- L114: why did you choose “10%” of the commercial SWE solution? While you applied 80% of the extracted solution.

- L121: please make the title of table 1 more specific and make the same for all tables and figures captions

- Table 1: Please change T0, T1, T2 to the correct treatments abbreviation and clearly explain the treatment in the table, it is misleading.

- L127: please change the title “2.4”

- L133: please explain how did apply the salt stress, it is not clear.

- L137: please delete this title and merge some paragraphs

- L167: please put the significance of “GCMS”

- L174: Since you have two factors, salinity and SWE, it is better to use two-way ANOVA instead of one-way.

Results

- L180: please delete the space between the value and % and change “to control” to “to the control”. Please correct the whole manuscript.

- L184: please change “to 80.6 from 20%” to “from 20 to 80.6%”. Please correct in the whole manuscript.

- Figure 1: please place the figure caption just after the figure and add the significance of all the abbreviations included in the figure. The font size should be enhanced.

- L192-194: this paragraph should be placed with the growth parameters.

- L194: please ovoid abbreviation in the table title and delete units. Please correct in all tables.

- Table 2: please use two decimals in all the values. Please correct in the whole section.

- Figure 3: please delete the photo, it is not significant. The same for Figures 4 and 7.

Discussion

- L320-322: please provide more detailed mechanisms related to the boosting effect of SWE on the germination process under salinity.

- L327 and 335: please italicize the scientific names.

Author Response

Response to reviewers

Manuscript ID agriculture-1973863

Title: Seaweed based biostimulant protect functionality of tomato plants grown under salt stress and improve nutraceutical as well as organoleptic traits of tomato fruits

Authors: Authors: Kanagaraj Muthu-Pandian Chanthini, Sengottayan Senthil-Nathan*, Ganesh-Subbaraja Pavithra, Arul-Selvaraj Asahel, Pauldurai Malarvizhi, Ponnusamy Murugan, Arulsoosairaj Deva--Andrews, Haridoss Sivanesh, Ramakrishnan Ramasubramanian, Ahmed Abdel-Megeed and Patcharin Krutmuang*

We thank the reviewers for their valuable suggestions for the improvement of the manuscript. We have carried out all the suggestions made by the reviewers that is made by using ‘track change’ and highlighted texts.

Reviewer 1

Q1

:

The English of this manuscript needs substantial improvements with a check of the punctuation.

Response

:

The language and punctuation errors were rectified.

Q2

:

Many choices made in M&M should be justified and many protocols are not well described.

Response

:

The methodology has been justified and detailed explanation of methods is added.

Q3

:

in the results section, please use % of variation instead of the recorded values

Response

:

% of variation are added .

Q4

:

Abstract

- L24: please add a space between the value and the unit. Please correct throughout the manuscript

- L30: please change “nutritional” to “nutraceutical”.

- Keywords: please change “marine algae” to “seaweed extract” and add salinity and tomato.

Response

:

Space between values and units are added throughout the manuscript.

“nutritional” was changed to “nutraceutical

Keywords were changed as suggested.

Q5

:

Introduction

- L45: at the beginning of the sentence you stated “According to recent studies” but you mention just one reference. Please correct.

- L57-58: please change “carotenoids, phenolics” to “carotenoids and phenolics”.

- L94: please change “SE” to “SWE”.

- L94-95: please rephrase.

Response

:

L45 – The reference corresponds to a review of literature. The sentence as changed ‘According to a recent studies literature review’

L57-58 – The sentence was changed as suggested.

L94: SE” was changed to “SWE”

L94-95 was rephrased.

Q6

L98: please write “Materials” without “s”

S was removed as suggested

Q7

- L101: please delete figure 1 and just put  the GPS coordinates of the site.

Figure 1 was deleted

Q8

- L103: please correct “((100 g/1 L- 1 h)”

The phrase was corrected.

Q9

- L107: Is it “150 ppm” or “150 mM”? please correct.

The error was corrected.

Q10

- L114: why did you choose “10%” of the commercial SWE solution? While you applied 80% of the extracted solution.

Commercial SWE is a purified form and as per direction of usage the concentration was made as 10%

Q11

- L121: please make the title of table 1 more specific and make the same for all tables and figures captions

The correction was carried out

Q12

- Table 1: Please change T0, T1, T2 to the correct treatments abbreviation and clearly explain the treatment in the table, it is misleading.

The abbreviation of treatments are mentioned below table 1 and explained in section 2.2. T0, T1…. was used to denote treatments to avoid overlapping in figure legends.

Q13

- L127: please change the title “2.4”

Title was changed

Q14

- L133: please explain how did apply the salt stress, it is not clear.

The salt stress was applied through irrigation water post transplantation and for 2 weeks at 2 day intervals – explained in section 2.4

Q15

- L137: please delete this title and merge some paragraphs

The methods were explained as different subheadings as to denote the effect of salt and SWE on various parameters of tomato plant growth and development.

Q16

- L167: please put the significance of “GCMS”

The significance was explained.

Q17

L174: Since you have two factors, salinity and SWE, it is better to use two-way ANOVA instead of one-way.

The treatments were analysed to study the effect of salinity of plant and the effect of seaweed extract on saline stressed plant. Only one variable was analysed at a time.

The authors thank the reviewer for the suggestion and will perform 2 way ANOVA in future experiments where 2 variables are co-analysed.

Q18

L180: please delete the space between the value and % and change “to control” to “to the control”. Please correct the whole manuscript.

The corrections were carried out.

Q19

L184: please change “to 80.6 from 20%” to “from 20 to 80.6%”. Please correct in the whole manuscript.

The correction was carried out

Q20

Figure 1: please place the figure caption just after the figure and add the significance of all the abbreviations included in the figure. The font size should be enhanced.

Figure caption was placed above figure as mentioned in the template. The treatments are already explained in table 1. Font size is enhanced.

Q21

- L192-194: this paragraph should be placed with the growth parameters.

SVI is a germination parameter that denoted the vigour of emerged seedling from treated seeds.

Q22

- L194: please ovoid abbreviation in the table title and delete units. Please correct in all tables.

The format of adding units in figure/table legends has been followed in previously published MDPI papers.

Q23

- Table 2: please use two decimals in all the values. Please correct in the whole section.

Changing decimal values will affect error and grouping categories of treatments.

Q24

- Figure 3: please delete the photo, it is not significant. The same for Figures 4 and 7.

We received a positive note on photos, figures and tables from other reviewers with slight modifications suggested by them that have been carried out. If the photos are deleted it would be taken by other reviewers as we went against their suggestion or would confuse them.

Q25

- L320-322: please provide more detailed mechanisms related to the boosting effect of SWE on the germination process under salinity.

The details have been added.

Q26

- L327 and 335: please italicize the scientific names

Scientific names have been italicised.

Reviewer 2 Report

1. Please shorten the title. It's too long.

2. The English expression of the full text needs further improvement.

3. Keyword needs to be re-selected. Remove words referring to broad meaning such as Abiotic stress, association, fruit yield, etc. No abbreviations.

4. In the introduction, it is suggested to reduce the introduction of tomato nutritional value and supplement the references of tomato salt stress. Add references on the effects of seaweed extract (SWE) on plant stress resistance, especially salt and alkaline tolerance.

5. It is recommended to move Figure 1 into the Supplementary.

6. It is not recommended to use hot water, etc. For such descriptions, please specify the temperature, and other parts should also use scientific language to describe the experimental methods as much as possible.

7. What is the basis for using 150 ppm sodium chloride for salt stress? If there is experimental evidence, it needs to be supplemented. Or add references.

8. "Watered with appropriate seen", please replace "appropriate" with exact data.

9. The experimental method needs to be described in more detail.

10. Please give clear calculation methods or formulas for each indicator.

11. It is suggested to replace table 1 with a text description.

12. The statement in 2.5.3 is inappropriate. The mitigation effect should be obtained from morphological observation rather than physiological and biochemical indicators, which are the possible internal mechanism of its mitigation effect.

13. Photosynthetic pigments should not be classified as nutritional quality.

14. For fruit organolepty, please add color, shape, color uniformity, etc.

15. Why only two concentrations are shown in Figure 4?

16. Figure 6, three units are used for different indicators ( mg/mg FW, μ g/mg protein, DW), need to recalculate and unify the unit, otherwise it cannot be compared.

17. All data in the figure shall be supplemented with significance and error labels.

18. Fig. 7, the fruit looks a little deformed. Please use the original photo.

19. Why only Figure 8 has trend lines added, but not others? Please keep the full text consistent.

20. The discussion is too complicated. Please focus on the main conclusions of the paper.

21. The key results of the experiment should be properly listed in the conclusion.

Author Response

Response to reviewers

Manuscript ID agriculture-1973863

Title: Seaweed based biostimulant protect functionality of tomato plants grown under salt stress and improve nutraceutical as well as organoleptic traits of tomato fruits

Authors: Authors: Kanagaraj Muthu-Pandian Chanthini, Sengottayan Senthil-Nathan*, Ganesh-Subbaraja Pavithra, Arul-Selvaraj Asahel, Pauldurai Malarvizhi, Ponnusamy Murugan, Arulsoosairaj Deva--Andrews, Haridoss Sivanesh, Ramakrishnan Ramasubramanian, Ahmed Abdel-Megeed and Patcharin Krutmuang*

We thank the reviewers for their valuable suggestions for the improvement of the manuscript. We have carried out all the suggestions made by the reviewers that is made by using ‘track change’ and highlighted texts. Reviewer 2

Q1

:

Please shorten the title. It's too long.

Response

:

Tate has been shortened

Q2

:

The English expression of the full text needs further improvement.

Response

:

Language has been corrected.

Q3

:

Keyword needs to be re-selected. Remove words referring to broad meaning such as Abiotic stress, association, fruit yield, etc. No abbreviations.

Response

:

Keywords have been changed.

Q4

:

In the introduction, it is suggested to reduce the introduction of tomato nutritional value and supplement the references of tomato salt stress. Add references on the effects of seaweed extract (SWE) on plant stress resistance, especially salt and alkaline tolerance

Response

:

The detail was added.

Q5

:

It is recommended to move Figure 1 into the Supplementary.

Response

:

Figure 1 has been deleted as per the suggestion of other reviewer.

Q6

:

It is not recommended to use hot water, etc. For such descriptions, please specify the temperature, and other parts should also use scientific language to describe the experimental methods as much as possible.

Response

:

The correction was carried out.

Q7

:

What is the basis for using 150 ppm sodium chloride for salt stress? If there is experimental evidence, it needs to be supplemented. Or add references.

Response

:

The experiment is substantiated with reference.

Q8

Watered with appropriate seen", please replace "appropriate" with exact data.

The data was added.

Q9

The experimental method needs to be described in more detail.

The description was added.

Q10

Please give clear calculation methods or formulas for each indicator.

Formulae has been added.

Q11

It is suggested to replace table 1 with a text description.

Table 1 has also been described as text

Q12

The statement in 2.5.3 is inappropriate. The mitigation effect should be obtained from morphological observation rather than physiological and biochemical indicators, which are the possible internal mechanism of its mitigation effect.

Mitigation effect is observed through morphological observation which was facilitated by physiological and biochemical changes that has been effected by  seaweed extract application in tomato plants under salt stress.

Q13

Photosynthetic pigments should not be classified as nutritional quality.

Increased photosynthetic pigment concentration indicates increased biomass accumulation encompassing carbohydrates and antioxidant pigments.

Q14

For fruit organolepty, please add color, shape, color uniformity, etc.

The color differences was added.

Q15

Why only two concentrations are shown in Figure 4?

Three concentrations of significantly different treatments are displayed.

Q16

Figure 6, three units are used for different indicators ( mg/mg FW, μ g/mg protein, DW), need to recalculate and unify the unit, otherwise it cannot be compared.

As mentioned in legend and figure figure 6A is different from 6B. They are not compared.

Q17

All data in the figure shall be supplemented with significance and error labels.

Significance and error labels have been supplied. For figures 6A and 7A the differences are explained in text.

Q18

Fig. 7, the fruit looks a little deformed. Please use the original photo

Photo given is original.

Q19

Why only Figure 8 has trend lines added, but not others? Please keep the full text consistent

Figure 8 compares yield and fruit vitamin C content which cannot be given in simple bar graph.

Q20

The discussion is too complicated. Please focus on the main conclusions of the paper.

Discussion cites obtained results and correlates with mechanisms cited by previously published papers.

Q21

The key results of the experiment should be properly listed in the conclusion.

The key results of the experiment are listed in the conclusion

Reviewer 3 Report

The manuscript “Seaweed based biostimulant protect functionality of tomato plants grown under salt stress and improve nutraceutical as well as organoleptic traits of tomato fruits” initiated with an interesting idea about the importance and potentiality of halotolerant marine macroalgal extract in alleviating the salt stress in tomato plants and promote the plant yield and yield related attributes.

However, the manuscript has some merit points, but not written well manner and it needs to clarify some major critical points to be re-checked and re-drafted prior to the Journal's revision and queries, as well as more refinement.

There are a lot of flaws, for example, spelling spaces. I strongly suggest you that spell out all abbreviations in the text the first time mentioned in the text and spell out all abbreviations in full form.

Cross-reference all of the citations in the text with the references in the reference section and make sure that all references have a corresponding citation within the text and vice versa.

English needs to be improved as lots of typographical errors, for which they must get the assistance of someone with a native command of the English language.

L22: Must revise the first line.

L23: (SWE) -> ? spell out

L44: “According to recent studies, the….. flavour [2].” there is only one reference and you mention that “studies” so add more references to justify the text.

L45: Provide the production of tomatoes.

L50: Provide the % of the content.

L55: Already highlighted “flavonoids”

L59: From L48-L58 what way is connected directly with this part “As a result, improving fruit quality through salt stress is a 59 critical subject for fresh foods with high nutritional content.”  

L79: Include some references.

L39: The introduction part needs to be the further clear and more accurate expression. (tomato, salinity problem, alleviation way, seaweed-based biostimulant role, goal of this investigation)

L102: SWE is very confusing, rewrite it properly in the full manuscript.   

L103: Check all superscripts

L106, 108: “Merck” add country name

L115: distilled water (PC) - ? spell out

L127: Provide more information in detail on the pot experiment.

L275: check “GCGCMS”

L291: the results part is well written.

L429: From GCMS analysis; try to include some text about the compound's role in the conclusions section.  

Author Response

Response to reviewers

Manuscript ID agriculture-1973863

Title: Seaweed based biostimulant protect functionality of tomato plants grown under salt stress and improve nutraceutical as well as organoleptic traits of tomato fruits

Authors: Authors: Kanagaraj Muthu-Pandian Chanthini, Sengottayan Senthil-Nathan*, Ganesh-Subbaraja Pavithra, Arul-Selvaraj Asahel, Pauldurai Malarvizhi, Ponnusamy Murugan, Arulsoosairaj Deva--Andrews, Haridoss Sivanesh, Ramakrishnan Ramasubramanian, Ahmed Abdel-Megeed and Patcharin Krutmuang*

Reviewer 3

Q1

:

There are a lot of flaws, for example, spelling spaces. I strongly suggest you that spell out all abbreviations in the text the first time mentioned in the text and spell out all abbreviations in full form.

Cross-reference all of the citations in the text with the references in the reference section and make sure that all references have a corresponding citation within the text and vice versa.

Response

:

Spelling spaces are included in template format of agriculture journal. It was not intended by the authors.

Q2

:

English needs to be improved as lots of typographical errors, for which they must get the assistance of someone with a native command of the English language.

Response

:

Typographical errors are rectified

Q3

:

: Must revise the first line.

Response

:

the line was revised

Q4

:

(SWE) -> ? spell out

Response

:

Seaweed extract (SWE) was spelled out.

Q5

:

According to recent studies, the….. flavour [2].” there is only one reference and you mention that “studies” so add more references to justify the text.

Response

:

The citation was a review article and the statement was a comprehensive description obtained from the review article.

Q6

:

L45: Provide the production of tomatoes

Response

:

The data was added

Q7

:

L50: Provide the % of the content.

Response

:

Q8

:

L55: Already highlighted “flavonoids”

Response

:

Q9

:

L59: From L48-L58 what way is connected directly with this part “As a result, improving fruit quality through salt stress is a 59 critical subject for fresh foods with high nutritional content.” 

Response

:

It is connected with the previous line that explains the nutritional quality of tomato plant.

“Fruits with high levels of bio active secondary metabolites such as carotenoids and phenolics, increase nutritional qualities and chemical composition, resulting in a high added value yield”

-        This line explains that nutritional quality, yield of fruits depend on bio active secondary metabolite levels.

“As a result, improving fruit quality through of plants subjected to salt stress is a critical subject for fresh foods with high nutritional content.”

-        And so improving fruit quality of plants affected by salt stress is important – as explained in the above line.

Q10

:

L79: Include some references

Response

:

References has been added.

Q9

:

L39: The introduction part needs to be the further clear and more accurate expression. (tomato, salinity problem, alleviation way, seaweed-based biostimulant role, goal of this investigation)

Response

:

The section has been modified.

Q9

:

L102: SWE is very confusing, rewrite it properly in the full manuscript.  

Response

:

The abbreviation has been explained in full when mentioned for the first time – S – Sea, W-weed and E – extract.

SWE for seaweed extract has been a widely used abbreviation since 1999 and many other studies followed through years. Below are some examples that SWE is neither a confusing nor new abbreviation.

1.     Shan, B.E., Yoshida, Y., Kuroda, E. and Yamashita, U., 1999. Brief communication immunomodulating activity of seaweed extract on human lymphocytes in vitro. International journal of immunopharmacology, 21(1), pp.59-70.

2.     Zhang, X., Ervin, E.H. and Schmidt, R.E., 2003. Physiological effects of liquid applications of a seaweed extract and a humic acid on creeping bentgrass. Journal of the American Society for Horticultural Science, 128(4), pp.492-496.

3.     Xu, C. and Leskovar, D.I., 2015. Effects of A. nodosum seaweed extracts on spinach growth, physiology and nutrition value under drought stress. Scientia Horticulturae, 183, pp.39-47.

4.     Abbas, M., Anwar, J., Zafar-ul-Hye, M., Iqbal Khan, R., Saleem, M., Rahi, A.A., Danish, S. and Datta, R., 2020. Effect of seaweed extract on productivity and quality attributes of four onion cultivars. Horticulturae, 6(2), p.28.

5.     Hussain, H.I., Kasinadhuni, N. and Arioli, T., 2021. The effect of seaweed extract on tomato plant growth, productivity and soil. Journal of Applied Phycology, 33(2), pp.1305-1314.

Q10

:

L103: Check all superscripts

Response

:

The expression is changed.

Q11

:

L106, 108: “Merck” add country name

Response

:

Country name is added.

Q12

:

L115: distilled water (PC) - ? spell out

Response

:

PC is spelled out.

Q13

:

L127: Provide more information in detail on the pot experiment.

Response

:

Details have been added.

Q13

:

L275: check “GCGCMS”

Response

:

Typographical error is rectified.

Q14

L291: the results part is well written.

Response

The authors thank the reviewer.

Q15

L429: From GCMS analysis; try to include some text about the compound's role in the conclusions section. 

Response

The conclusion section has been modified.

Round 2

Author Response

Q1

Please mention this in the text

Response

The description was added in text

Q2

Since you have a combined effect of Salt and SWE, you have two factors. Please perform two-way ANOVA

Response

Combined effect was not studied as a single unit. We used separate treatments to study effect of salt (T5) and effect of Seaweed extract on salt stressed plants (T7).

Q3

I checked the template where Fig. captions are placed below Figs. Please correct and add the significance of all the abbreviations, we do not have to go back to table 1 to know the significance.

Please correct the numbering of Figures.

Response

The modifications were carried out.

Q4

Nothing will change, you should just keep two decimals or three, the same number of decimals for all values.

Response

The modifications were carried out

Q5

From a scientific point of view, it is necessary to put several repetitions for the same treatment to be significant, so I do not agree with these photos

Response

We have already mentioned in section 2.6 that all experiments were replicated 5 times.

Reviewer 2 Report

The author made corresponding revisions according to the comments of the reviewers, and I think the paper has basically reached the publishing level.

Author Response

Since the expert mentioned only minor spell check we have carried out all

Reviewer 3 Report

Significantly improved, yet the author completely ignored several critics' suggestions of reviewers. However, I advise you to undertake a critical, scientific revision of the text.

L36, 42, 81, 261: "./;;" check; rectify all the errors in the manuscript.  Before submitting the revised one, check the manuscript critically.   

Figures 6 A and 7 A: must be revised, re-draw the figures using bar by / line graph or another way.

Some of the figures must be removed or changed because they are not presented scientifically in a pleasant manner.

Table 5: "4H-Pyran-4-one,2,3-dihydro-3,5-... " check "...."

Author Response

Q1

L36, 42, 81, 261: "./;;" check; rectify all the errors in the manuscript.  Before submitting the revised one, check the manuscript critically

Response

The errors have been rectified.

Q2

Figures 6 A and 7 A: must be revised, re-draw the figures using bar by / line graph or another way.

Response

We would like to keep the figures in the original way as no scientific flaw was quoted.

Q3

Some of the figures must be removed or changed because they are not presented scientifically in a pleasant manner

Response

We would keep  the figures as previous reviewers did not raise any concern on the scientific genuinity of the figures.

Q4

Table 5: "4H-Pyran-4-one,2,3-dihydro-3,5-... " check "...."

Response

The correction was carried out